# Nanoparticles (PLGA and Chitosan)-Entrapped ADP-Ribosylation Factor 1 of *Haemonchus contortus* Enhances the Immune Responses in ICR Mice

**DOI:** 10.3390/vaccines8040726

**Published:** 2020-12-02

**Authors:** Muhammad Waqqas Hasan, Muhammad Haseeb, Muhammad Ehsan, Javaid Ali Gadahi, Muhammad Ali-ul-Husnain Naqvi, Qiang Qiang Wang, Xinchao Liu, Shakeel Ahmed Lakho, Ruofeng Yan, Lixin Xu, Xiaokai Song, Xiangrui Li

**Affiliations:** MOE Joint International Research Laboratory of Animal Health and Food Safety, College of Veterinary Medicine, Nanjing Agricultural University, Nanjing 210095, China; 2015207037@njau.edu.cn (M.W.H.); 2016207041@njau.edu.cn (M.H.); mehsan124@gmail.com (M.E.); drgadahi@yahoo.com (J.A.G.); 2017207047@njau.edu.cn (M.A.-u.-H.N.); 2016107053@naju.edu.cn (Q.Q.W.); 2015207017@njau.edu.cn (X.L.); 2016207025@njau.edu.cn (S.A.L.); yanruofeng@njau.edu.cn (R.Y.); xulixin@njau.edu.cn (L.X.); songxiaokai@njau.edu.cn (X.S.)

**Keywords:** *Haemonchus contortus*, rHcARF1, PLGA, CS, nanoparticle, ICR mice

## Abstract

ADP-ribosylation factor 1 (HcARF1) is one of the *Haemonchus contortus* (*H. contortus*) excretory/secretory proteins involved in modulating the immune response of goat peripheral blood mononuclear cells (PBMC). Here, we evaluated the immunogenic potential of recombinant HcARF1 (rHcARF1) against *H. contortus* infection in Institute of Cancer Research (ICR) mice. Briefly, rHcARF1 was entrapped in poly (D, L-lactide-co-glycolide) (PLGA) and chitosan (CS) nanoparticles (NP) and injected into mice as a vaccine. Fifty-six ICR mice were assigned randomly into seven groups, with eight animals in each group, and they were vaccinated subcutaneously. At the end of the experiment (14th day), the blood and the spleen were collected from euthanized mice to detect lymphocyte proliferation, cytokine analysis, and the production of antigen-specific antibodies. Scanning electron microscope was used to determine the size, morphology, and zeta potential of nanoparticles. Flow cytometry was performed, which presented the increase percentages of CD4^+^ T cells (CD3e^+^CD4^+^), CD8^+^ T cells (CD3e^+^CD8^+^) and dendritic cells (CD11c^+^CD83^+^, CD11c^+^CD86^+^) in mice vaccinated with rHcARF1+PLGA NP. Immunoassay analysis show raised humoral (Immunoglobulin (Ig)G1, IgG2a, IgM) and cell-mediated immune response (Interleukin (IL)-4, IL-12, and IL-17, and Interferon (IFN)-γ) induced by rHcARF1+PLGA NP. Experimental groups that were treated with the antigen-loaded NP yield higher lymphocyte proliferation than the control groups. Based on these results, we could propose that the rHcARF1 encapsulated in NP could stimulate a strong immune response in mice rather than administering alone against the infection of *H. contortus*.

## 1. Introduction

*H. contortus* is one of the gastrointestinal nematodes that infect the sheep and goat through feces and cause the disease termed as “haemonchosis” [1]. The infestation of *H. contortus* affects thousands of sheep and goats annually, and substantial economic losses to farmers are reported [2]. This parasitic nematode goes into the stomach of the host animal from the herbage while grazing. It causes the infection that leads to anemia, dehydration, and protein loss in adult animals, and it may cause death in young lambs [3].

*H. contortus* releases ARF1 during various stages of infections in goats [4,5]. ARF1 is part of the Ras-related small GTPases family. Primarily, they are involved in the regulation of vesicular trafficking [6]. Moreover, it is an essential regulator of the biological process induced by epidermal growth factor [7,8,9]. ADP-ribosylation factor 6 (ARF6) is another variant of the ARF proteins family, which mainly controls the neuronal morphogenesis processes and membrane trafficking [10]. Small GTPases found in shrimp participate in inducing antiviral immunity by regulating phagocytosis [11,12]. ARF1 and ARF2 might play essential roles in the innate immune response against white spot syndrome virus infection [13]. ARFs have been identified in several plant species, including Arabidopsis, barley, carrot, maize, potato, rice, tomato, tobacco, and wheat [14]. The replication of the plant RNA virus also needed the participation of Arf1 [15]. Moreover, HcARF1 could stimulate the immune response of goat immune cells in vitro [5]. In various investigations, different antigens isolated from *H. contortus* were used in designing vaccines that showed significant protection in the host animal against this parasite [16,17,18,19].

Vaccines consist of adjuvants that have a critical role as stabilizing compounds, and without them, vaccines are not effectively immunogenic [20]. Freund designed the most effective and well-known adjuvant as complete Freund’s adjuvant (CFA) [21]. CFA is considered a gold standard due to its highly specific and good immunogenicity [22]. However, because of severe reactions at the injection sites and the possible residues in meat, the use of CFA is risky in farm animals. Therefore, a potent and well-tolerated adjuvant system has become the prerequisite in developing vaccines for domesticated animals [23]. Moreover, an adjuvant should efficiently deliver antigen-presenting cells (APC), including T cells and dendritic cells (DC), to exert a robust immunogenic response [24].

In the medical field, nanotechnology offers an excellent opportunity to design biodegradable nanoparticles (NP) varying in size, composition, surface properties, and shape for their application [25]. Numerous approved nano-sized vaccine and drug delivery systems have highlighted that the breakaway is preventing and treating infectious diseases [26]. Several scientists have previously reported the potent immunological effects of poly (D, L-lactide-co-glycolide) (PLGA) and CS using some model antigens [27,28,29].

To date, very little information about the isolated antigens of *H. contortus* encapsulated in polymeric NP used as an immunogenic agent is available. Therefore, we employed biodegradable polymers (PLGA and CS) as the adjuvants that carried an antigen of *H. contortus*. Nanomaterials (PLGA and CS) and encapsulated HcARF1 elicited robust humoral and cell-mediated immune responses and conferred significant protection against a helminth challenge. Thus, these antigen-loaded NP can now be explored as potential vaccine candidates.

## 2. Materials and Methods

### 2.1. Ethics Statement

The instructions of the Animal Ethics Committee, Nanjing Agricultural University, China, were followed during the study. All laboratory and animal experiments were performed by following the Animal Welfare Council of China’s guidelines. The Science and Technology Agency approved all experimental protocols of Jiangsu Province. The approval ID is SYXK (SU) 2010-0005.

### 2.2. Materials

PLGA (lactic acid: glycolide 65:35, molecular weight (Mw) = 40,000–75,000), chitosan (Mw = 50,000–190,000 Da), poly vinyl alcohol (PVA, Mw = 31,000–50,000), concanavalin A, and complete Freund’s adjuvant were purchased from Sigma-Aldrich (St. Louis, MO, USA).

We purchased the Sodium tripolyphosphate (Mw = 367.86) from Aladdin and Enhanced Cell Counting Kit-8 from Beyotime. Likewise, the BCA Protein Assay Kit was acquired from CW Biotech, China.

The Roswell Park Memorial Institute medium (RPMI 1640), heat-inactivated fetal bovine serum (FBS), and penicillin–streptomycin solutions were purchased Gibco. Antibodies PE rat anti-mouse CD86 (clone: GL1), PE rat anti-mouse CD83 (clone: Michel-19), APC hamster anti-mouse CD11c (clone: HL3), APC hamster anti-mouse CD3e (clone: 145-2C11), FITC rat anti-mouse CD4 (clone: RM4-5), and FITC rat anti-mouse CD8α (clone: 53-6.7) were purchased from Bio-legend. The purified recombinant proteins of HcARF1 and pET-32a expressed in *E. coli* [5] were obtained from Molecular Parasitology and Immunology laboratory at Nanjing Agricultural University.

### 2.3. Animals

Fifty-six specific pathogen-free (SPF) female Institute of Cancer Research (ICR) mice (age = two-weeks, body weight= 18–20 g) were purchased from the Experimental Animal Center of Jiangsu, China (Qualified Certificate: SCXK 2017-0001). Female mice are generally tested in toxicology, neurobiology, oncology, infection, pharmacology studies, and more reliable data can be obtained. Mice were housed in a specific pathogen-free environment and were given ad libitum access to sterilized food and water.

### 2.4. Preparation of Recombinant Protein of H. contortus (rHcARF1)

The plasmid to express the recombinant protein HcARF1 were constructed previously, and the recombinant proteins were expressed and purified as described earlier [5]. In brief, the recombinant plasmids pET-32a (+) with HcARF1 were shifted into BL21 (*E. coli*, DE3) and then cultured in Luria Bertini (LB) medium containing ampicillin (100 μg/mL), induced with isopropyl-β-D-thiogalactoside (IPTG) at a concentration of 1 mM for four h) at 37 °C. The cell pellet obtained from culture was suspended in 100 mL of wash buffer (10 mM Tris-HCl (pH 7.4) containing 0.5 M NaCl, five mM 2-mercaptoethanol, 1 mM EDTA, and 1 mM PSMF) and then sonicated for 15 min on ice. Afterward, the sonicate was supplemented with 1% (*w*/*v*) Triton X-100 and then stirred for 30 min at 4 °C, followed by centrifugation. The proteins were purified by affinity chromatography over Ni-NTA resin using an imidazole gradient elution according to the manufacturer’s recommendations (His-Bind^®^ Resin Chromatography kit, Novagen). Next, the proteins were dialyzed in phosphate-buffered saline (PBS, pH 7.4) to remove imidazole. The purity of the rHcARF1 protein was determined by 12% SDS-PAGE with staining by Coomassie blue and quantified by the Bradford method [30]. Endotoxins were removed using the Toxin-Eraser™ Endotoxin Removal kit (GeneScript, Piscataway, NJ, USA) from the recombinant proteins.

### 2.5. Preparation of Antigen-Loaded PLGA and CS NP

In the preparation of PLGA nanoparticles (NP), PVA is a crucial ingredient and played a vital role as a surfactant [31]. Before starting the trial of NP, we determine the working concentration of PVA (6%).

Polymeric PLGA NP were prepared according to the double emulsion (w/o/w) solvent evaporation method [32], under sterile conditions. Briefly, a protein of rHcARF1 (1 mg/mL) was dissolved in 6% (*w*/*v*) PVA solution to form the inner aqueous phase. Then, we dissolved 5% (*w*/*v*) PLGA in methylene chloride (50 mg PLGA in 1 mL methylene chloride). Two stages were combined to form w/o emulsion and sonicated (40 w, 5 s, 5 s) using an ultrasonic machine (JY92-IIN, NingBo Scientz Biotechnology, Ningbo, China) for 4 min in an ice bath. Next, w/o emulsion was transferred to an external aqueous phase containing 6% PVA dissolved in deionized water and was sonicated again (40 w, 5 s, 5 s) for 4 min to acquire final w/o/w emulsion. The organic solvent was vaporized by keeping it at room temperature (RT) for 4 h under magnetic stirring. Finally, NP was formed due to polymer precipitation. The solution of PLGA NP was centrifuged at 20,000× *g* for 40 min at 4 °C. After collecting the supernatant, the precipitated NP were washed three times with ultrapure water by centrifugation.

Recombinant HcARF1-loaded chitosan (CS) NP was prepared according to the ionic gelation method [33,34,35,36]. In brief, 200 mg CS was dissolved in a final volume (100 mL) of 1% acetic acid and stirred on a magnetic stirrer at RT for 30 min. The 2.0N NaOH was used to adjust the solution′s pH. The antigen was added to CS solution (5.7 mL) in drop by drop fashion. Next, 4 mL of sodium tri-polyphosphate was mixed in the CS-antigen solution with continuous magnetic stirring at RT. Then, the mixture was sonicated for 4 min (50 w, 5 s, 5 s). The formulated rHcARF1 NP were centrifuged at 20,000× *g* for 40 min. As described above, precipitated NP were washed three times with ultrapure water by centrifugation.

Afterward, PLGA and CS NP were frozen in a freeze-drier for 24 h and then stored in a −80° refrigerator before used.

The empty PLGA and CS NP were prepared by following the same method described before; however, the antigen did not mix with biopolymers.

### 2.6. Characterization of Antigen-Loaded NP

#### 2.6.1. Loading Capacity (LC), Encapsulation Efficiency (EE), and Cumulative Release

The obtained supernatants collected after washing of NP (PLGA, CS) were used to determine protein loading capacity (LC) and encapsulation efficiency (EE) using the BCA protein assay kit by the following equations [37,38].
EE = (total protein − unbound protein)/total protein × 100%
LC = loaded protein/total mass of nano-vaccine × 100%

The release of recombinant HcARF1 protein from PLGA and CS nanoparticles was assayed in vitro performed by monitoring changes of the free recombinant HcARF1 in solution. Three milligrams of lyophilized nanoparticles were dispersed in 150 μL of sterile PBS (0.1 M, pH7.4) and placed in a shaker bath (37 °C, 120 rpm). At specific time intervals (0, 1, 3, 5, 7, 9, 11, 13, 15th days), the suspension was centrifuged at 12,000 rpm for 15 min, and 60 μL of supernatant was removed and immediately replaced with the same volume of fresh PBS. A Micro-BCA Protein Assay Kit determined the concentration of free rHcARF1 protein in the supernatants. All analyses were performed in triplicate [39].

#### 2.6.2. Scanning Electron Microscopy for Morphology, Size, and Zeta Potential Measurements of Antigen-Loaded NP

The morphology and size of rHcARF1+PLGA and rHcARF1+CS nanoparticles were observed using a cold field emission (JEOL IT-100, S-4800 N) scanning electron microscope (SEM, Japan). The powder form of antigen-loaded NP was filled into aluminum stubs and coated with platinum before examination under the microscope.

The zeta potential of NP was measured using a zeta potential analyzer (Zeta plus, Brookhaven Instruments Co, New York, NY, USA). All zeta estimations were determined at 25 °C in an electric field of 11.00 V/cm [39].

#### 2.6.3. Integrity of the Antigen–NP Complex

Antigen-loaded NP suspensions were digested at 95 °C for 20 min and loaded at RT into a gel (20 μL for samples and five μL for MW markers (Thermo Fisher Scientific, 10–180 kDa). The electrophoresis was performed at a constant voltage of 120 V for 90 min, using a Bio-Rad 300 power pack (Bio-Rad, Hercules, CA, USA). SDS-PAGE gels were further stained with 0.025% Coomassie Brilliant Blue to reveal protein bands [40].

### 2.7. Immunization Protocol

Fifty-six female ICR mice were randomly divided into seven groups of eight animals. We set four groups of mice as control groups, including PBS (blank control) pET-32a protein, CS, and PLGA. Three groups of mice, rHcARF1, rHcARF1+CS, and rHcARF1+PLGA, received treatments. The vaccine containing protein was injected subcutaneously into multiple mice following the earlier [41]. The vaccine was injected with a needle (28 gauge) at 0 days, and all mice were euthanized on the 14th day.

The different groups of animals receiving the vaccine are summarized in Table 1.

### 2.8. Observation of Gross lesions

All mice were regularly checked for any findings related to nervous signs and clinical symptoms of necropsy lesions during the whole experiment to evaluate vaccine safety.

### 2.9. Antibody Assays

Mice blood was collected before sacrificed on the 14th day, as described in a previous study [42]. The levels of antigen-specific antibodies (IgG1, IgG2a, IgM) produced by mice in sera samples were determined using commercially available mouse ELISA kits by following the manufacture’s instruction (HengYuan, Shanghai, China). In brief, rHcARF1 (20 μg/mL) was used to coat the ninety-six well microtiter plate overnight at 4 °C. After washing three times with 0.01 M PBS containing 0.05% Tween-20 (PBST), the wells were blocked with 5% non-fat dry skim milk powder (SMP) in PBST for 2 h at 37 °C, and 100 μL of the serum samples diluted 1:50 in PBST-5% SMP were added for one h at 37 °C. Following washing with PBST, wells were incubated with HRP conjugated anti-mouse IgG1, IgG2a, and IgM (diluted to 1:3000 in blocking buffer, Sigma Aldrich (St. Louis, MO, USA) antibodies for one h at 37 °C to determine antibody levels and isotype analysis. Tetramethyl benzidine (TMB) (Sigma) substrate was used to develop colors. Finally, a spectrophotometer was used to observe the results at an absorbance of 450 nm. All serum samples were determined with three replicates.

### 2.10. Antigen-Specific Cytokines Determination by ELISA

The commercial double antibody sandwich ELISA kits (HengYuan, Shanghai, China) were used to evaluate cytokines’ level (IL-4, IL-12, IL-17, IFN-γ, and TGF-β) produced in the sera of all experimental mice, according to the manufacturer’s instruction.

### 2.11. Lymphocyte Proliferation Assay

Lymphocyte proliferation assay was used to evaluate rHcARF1-specific lymphocyte activation [36]. On the 14th day, mice were euthanized to isolate spleen lymphocytes in the presence of antigen-presenting cells (APC) using Mouse Spleen Lymphocyte Isolation Kit (TBD, Tianjin, China) under sterilized conditions. Briefly, the RPMI-1640 culture medium (CM) was used to adjust the cell concentration (1 × 10^7^ cells/mL) and then cultured in 6-wells cell culture plates overnight. Next, cell supernatants (T and B cells) were collected and adjusted the concentration to 1 × 10^6^ cells/mL. Then, 1 × 10^6^ cells in 100 μL of CM (supplemented with 10% heat-inactivated fetal calf serum, 100 U/mL penicillin, and 100 mg/mL streptomycin) were incubated in each well of round-bottom 96-well culture plates and incubated again for the next 72 h under 5% CO_2_ at 37 °C. Next, the corresponding samples were re-stimulated by recombinant antigen (20 μg/mL). The supplemented samples and concanavalin A (ConA) was conducted and considered as a blank and positive control [37]. Lymphocyte proliferation induced by antigen (rHcARF1) was then determined by the Enhanced Cell Counting Kit-8 (Beyotime) according to the manufacturer’s instructions. The absorbance was measured using a microliter ELISA reader (Thermo Scientific Multiskan FC) at a wavelength of 450 nm (A_450_ value). The values were expressed as the stimulation index (SI), based on the following equation [43]:SI (100%) = A_t/_A_c_.

In the above equation, A_t_ indicates the mean A_450_ value of the test group, and A_c_ shows a blank control group.

### 2.12. Flow Cytometry

For the percentages of CD4^+^ T cells, washed cells were labeled with anti-CD3e-APC and anti-CD4-FITC. Similarly, the quantities of CD8^+^ T cells stained with anti-CD3e-APC and anti-CD8α-FITC were analyzed using fluorescence-activated cell sorting (FACS) Caliber.

For the DC maturation, the Mouse Spleen Lymphocyte Isolation kit was used to isolate the splenic cells described before, and a mixture of cells was cultured overnight. The supernatant of cells was discarded, and the plate was washed using PBS. The attached cells were pipetted gently and repeatedly. Following centrifuge collection and washing, cells were stained with anti-CD11c-APC, anti-CD83-PE, and anti-CD86-PE for the percentages of CD83^+^ and CD86^+^. Subsequently, FACS Caliber was used to performed flow cytometry.

### 2.13. Statistical Analysis

All experiments were implemented in triplicate, and data were presented as mean ± SEM. The one-way analysis of variance (ANOVA) test was employed by using the GraphPad Premier 7.0 software package (GraphPad Prism, San Diego, California, USA); to clarify the significant difference in the data, it was set to *p* < 0.05, *p* < 0.01, and *p* < 0.001. FACS data analysis was conducted using Flow Jo version 10 software.

## 3. Results

### 3.1. Characterization of Antigen-Loaded NP

#### 3.1.1. Shape and Zeta Potential of NP

The SEM result showed that appearances of NP were smooth and spherical in PLGA and CS NP (Figure 1A,B). Moreover, it revealed that the size of rHcARF1+PLGA NP was 100 ± 10 nm (Figure 1C) and of rHcARF1+CS NP was 260 ± 35 nm (Figure 1D). Furthermore, the zeta potential values of both NP were 24 ± 1.8 and 18 ± 2.2 mV respectively (Figure 1E,F). Our findings suggested that the positively charged NP interacts with negatively charged cell membranes and enhances NP uptake by antigen-presentation cells, specifically DC [38].

Figure 1A,B shows an SEM picture of rHcARF1+PLGA and rHcARF1+CS. The structure of nanoparticles was smooth (viewed at 10,000× magnification). Figure 1C–F shows the size distribution and zeta potential of rHcARF1+PLGA and rHcARF1+CS.

#### 3.1.2. Antigen Integrity Was Not Affected by Vaccine Formulation Procedures

The antigen-loaded nanoparticles ran in SDS-PAGE (12% separating gel) to check the binding and integrity of rHcARF1 with PLGA and CS NP. It clearly showed that Mw of the protein was not affected by the NP formulation process, and both NP appeared bands with a remarkable size of about 38 kDa, while faded bands showed the excess protein of rHcARF1 (Figure 2A).

#### 3.1.3. Cumulative Release Assay of Antigen

The protein release kinetics were determined in vitro as a cumulative release assay shown in Figure 2B. About 83% and 68% of antigen was released from PLGA and CS NP respectively after 14 days. According to the results, the nanoparticles may be effective antigen delivery vehicles with small particle size and stability [38].

SDS-PAGE (12% separating gel) ran to examine the binding and integrity of rHcARF1 with PLGA and CS nanoparticles. Lane M: standard protein molecular weight marker. Lane 1: PLGA NP with unbounded rHcARF1. Lane 2: PLGA NP with bounded rHcARF1. Lane 3: CS NP with bounded rHcARF1. Lane 4: CS NPs with unbounded rHcARF1 (Figure 2A). In vitro release profile of antigen from PLGA and CS NP at pH 7.4 at 37 °C for 14 days, calculated as a percentage release (Figure 2B).

#### 3.1.4. The Loading Capacity (LC) and Encapsulation Efficiency (EE) of Antigen-Loaded NP

A BCA protein assay kit determined free amounts of rHcARF1 protein entrapped in nanoparticles. According to results, 81.3% and 76% of rHcARF1 were encapsulated (EE) in PLGA and CS NP, respectively, and almost 28.8% and 40.6% of protein were loaded (LC) by PLGA and CS NP (Table 2).

### 3.2. Safety Assessment of Vaccination

Mice immunized with the vaccine did not show any nervous signs, clinical symptoms, or necropsy lesions during the experimental trial. However, further studies are needed to ensure a high level of safety.

### 3.3. Evaluation of Antigen-Specific Serum Antibodies Induced by Nanovaccine

The levels of IgG1, IgG2a, and IgM in sera of all groups of mice were determined by ELISA, as illustrated in Figure 3. The data indicate the significant effect of antigen and the antigen–NP complex on the stimulation of antigen-specific antibody immune response, resulting from their ability to promote antigen uptake and cross-presentation [38]. The results demonstrated that IgG1 levels in experimental groups (rHcARF1, rHcARF1+CS, and rHcARF1+PLGA) were significantly higher as compared to the PBS group and also higher than those in other control groups (pET-32a protein, CS, and PLGA) (* *p* < 0.05, ** *p* < 0.01, *** *p* < 0.001). While the rHcARF1-PLGA NP produced significantly increased IgG1 as compared to rHcARF1 and rHcARF1+CS groups (* *p* < 0.05, ** *p* < 0.01) (Figure 3A). Similarly, mice immunized with vaccine formulations and antigen alone secreted intense levels of IgG2a and IgM when compared with PBS and other control groups (* *p* < 0.05, ** *p* < 0.01, *** *p* < 0.001). Similar to IgG1, the rHcARF1+PLGA produced significantly increased quantities of IgM as compared to the rHcARF1+CS and rHcARF1 groups (** *p* < 0.01) (Figure 3C).

### 3.4. Modulation of rHcARF1-Specific Cytokine Production

The results showed a high level of IL-12, and IL-4 was produced by the treatment groups (rHcARF1, rHcARF1+CS, and rHcARF1+PLGA) compared to PBS and other control groups (pET-32a protein, CS, and PLGA) (* *p* < 0.05, ** *p* < 0.01, *** *p* < 0.001). Moreover, mice immunized with rHcARF1+PLGA produced more IL-4 than rHcARF1+CS and rHcARF1 groups (* *p* < 0.05) (Figure 4A,B).

According to the obtained results, the levels of IL-17 in treatment groups were increased compared with PBS and other control groups (** *p* < 0.01, *** *p* < 0.001). However, no difference was noticed when compared to the IL-17 production by rHcARF1 and rHcARF1+CS groups. Moreover, rHcARF1+PLGA induced more IL-17 as compared to rHcARF1+CS and rHcARF1 groups (* *p* < 0.05) (Figure 4C).

Significantly decreased IFN-γ was observed in all experimental groups compared with PBS, pET-32a, CS, and PLGA groups (** *p* < 0.01). Interestingly, the low production of IFN-γ by the treatment groups was evident compared with CS and PLGA due to the suppressing effect of antigen (Figure 4D).

TGF-β secretion levels did not change considerably among all experimental groups (Figure 4E).

### 3.5. Lymphocyte Proliferation Induced by Antigen and Antigen-Loaded Nanoparticles

Lymphocyte proliferation assays are widely used to assess cell-mediated immunity [44]. Lymphocyte activation occurs when lymphocytes (B cells or T cells) are triggered through antigen-specific receptors on their cell surface and they proliferate and differentiate into specialized effector lymphocytes. Therefore, we isolated splenic lymphocytes from all groups of mice on the 14th day, and their proliferative responses specific to rHcARF1 were calculated and expressed as SI values (Figure 5. According to the results, positive control (ConA), rHcARF1 + CS, and rHcARF1 + PLGA groups produced the most significant proliferation when compared with blank control (PBS), pET-32a protein, and empty NP (CS and PLGA) (** *p* < 0.01, *** *p* < 0.001). The antigen (rHcARF1) group also displayed higher proliferation than the PBS and other control groups (* *p* < 0.05).

### 3.6. Antigen and Antigen-Loaded Nanovaccine Promoted CD4^+^ and CD8^+^ T Cells Stimulation

T cells (CD4^+^CD8^+^) are highly activated cells exhibiting an effector memory phenotype. Previous studies have attributed regulatory properties to CD4^+^CD8^+^ T lymphocytes in animal models [45,46] and enhanced the production of Th2 associated cytokines. Hence, we evaluated the percentages of CD4^+^ and CD8^+^ T cells in each group by flow cytometry (Figure 6). As the results showed, significantly increased percentages of CD3^+^CD4^+^ cells were observed in all treatment groups than control groups (*** *p* < 0.001). rHcARF1+PLGA NPs group produced a non-significant increase in CD3^+^CD4^+^ cells when compared with rHcARF1, and rHcARF1+CS groups.

The results showed that the treatment groups produced significantly higher CD3^+^CD8^+^ cells than control groups (*** *p* < 0.0001). Moreover, compared to rHcARF1 and rHcARF1+CS groups, the rHcARF1+PLGA group generated a higher percentage of CD3^+^CD8^+^ cells (Figure 6B,D).

### 3.7. Antigen and Antigen-Loaded Nanovaccine Induced DC Phenotypes

DC constitutes only 1% of all peripheral blood mononuclear cells (PBMC) of a living body [47]; however, they exert potent regulatory effects on both the innate and adaptive immune systems. Thus, the expressions of CD11c^+^CD83^+^ and CD11c^+^CD86^+^ on splenic DC were compared in all experimental groups (Figure 7). The results displayed that the treatment groups showed greater augmentation in the percentages of CD11c^+^CD83^+^ cells compared with PBS and other control groups (*** *p* < 0.001). Meanwhile, the rHcARF1+PLGA group showed highest levels of CD11c^+^CD83^+^ cells when compared with the rHcARF1 and rHcARF1+CS groups (* *p* < 0.05) (Figure 7A,C).

The results showed that percentages of CD11c^+^CD86^+^ cells in the treatment groups were enhanced compared with the control groups (****p* < 0.001). The percentages of CD11c^+^CD86^+^ cells in the rHcARF1+PLGA group were increased significantly when compared with the rHcARF1 group (* *p* < 0.05). Furthermore, the value of CD11c^+^CD86^+^ in the rHcARF1+PLGA group was non-significantly high compared with the rHcARF1+CS group (Figure 7B,D).

## 4. Discussion

ADP-ribosylation factor 1 isolated from *H. contortus*, and it was reported that this protein is actively bound to goat PBMC in different larval stages of this helminth [4]. Previously, this molecule stimulated goat PBMC and regulated other functions of the immune system [5]. PLGA and CS have attracted much attention in recent years due to their clinically proven biocompatibility and adjuvant activity while running different immunization protocols [48,49,50,51]. Biodegradable particles with entrapped antigens, such as proteins, peptide, or DNA, have been shown to possess significant potential as vaccine delivery systems [52]. Earlier studies showed that PLGA and CS played their roles as potential nanocarriers to deliver antigens in animal models and cause an effective immune response [53,54]. Nanomaterials have many desirable properties for immunomodulation, as NP have the characteristic ability to passively target APC by mimicking the size and shape of an invading pathogen, increasing antigen uptake, processing, and cross-presentation [55]. Moreover, NP can be specifically designed to be recognized and promote the sustained delivery of antigens to APC and to further modulate intracellular signaling pathways toward the stimulation of long-lasting specific immune responses and, therefore, increase overall vaccine efficiency [56]. The purpose of this research was to evaluate the ability of rHcARF1 encapsulated in PLGA and CS NP that initiated protective immune responses in mice.

A modified technique named a double emulsion (w/o/w) was used to encapsulate biological drugs, such as peptides, proteins, and nucleic acids in NP [57]. NP preparation methods affected the yield and encapsulation efficiency (EE) [58] with the loading capacity (LC) of finally synthesized NP. Mostly, NP have a comparable size to pathogens and are consequently picked up efficiently by APC to induce immune response [59,60]. Usually, the size of NP is in the range of 100–250 nm [61]. In the current research, a high EE of NP was achieved (PLGA = 81.33, CS = 76) (Table 2). The high value of EE and the right size of NP (PLGA and CS) established that the vaccine formulations possess great physicochemical features. They would be highly suitable for further study. However, one of the significant disadvantages of PLGA-based NP relates to the poor LC [62]. In this study, a relatively low LC of rHcARF1+PLGA NP was obtained (Table 2) compared to rHcARF1+CS NP. Therefore, this needs to be further elucidated.

Among six mammalian Arfs (ADP-ribosylation factors), Arf4 and Arf5 played essential roles in the secretion of dengue virus [62]. Arfs have been identified in several plant species, including Arabidopsis, rice, tomato, potato, maize, carrot, wheat, tobacco, and barley [14]. The role of small GTPase Arfs in NP is seldom investigated. In a study of Arf6, the normal rat kidney (NRK) cells and Hela cells co-culture system was employed, and the TiO_2_ NPs transfer was observed. The authors found that the small GTPase Arf6 facilitates the intercellular transfer of smaller NP and endosomes [63]. Several studies have used antigens isolated from veterinary interest pathogens, showing increasing curiosity in drug delivery through NP entrapped antigens [20,43,64]. Therefore, we cloned and expressed rHcARF1 in *E. coli* prokaryotic expression system in a previous study [5], encapsulated rHcARF1 in PLGA and CS NP, and then vaccinated the mice to evaluate its immunogenicity against *H. contortus.* This research reported a robust immune response in vaccinated mice suggesting the adjuvant nature of polymeric NP (PLGA and CS) and antigenic role of rHcARF1.

Activated T helper 2 (Th2) cells secreted IL-4, which majorly holds the biological effects such as the proliferation and differentiation of B cells to produce antibodies and further promote T cells (CD4^+^); these are essential in controlling humoral and adaptive immunity [65]. Typically, against extracellular parasites, including helminths, IL-4 and IL-2 are triggered, and their effector cytokines could promote T cells and natural killer (NK) cells to secrete IFN-γ indirectly by stimulating dendritic cells [66]. IL-12 is essential for fighting infectious diseases produced primarily by monocytes, macrophages, and other APC such as DC or T cells [67]. We observed that within the splenic DC compartment of mice, the CD8α subset could be induced to secrete much higher levels of IL-12 [68], and interestingly, the presence of IL-4 in culture could enhance IL-12 from CD8α subset. Moreover, in mice, splenic CD8α DCs (30, 31) induce T helper 1 (Th1) and Th2 responses by the differential production of IL-12 [68,69]. In a recent study, the effect of rHcARF1 on the production of IL-4, IL-12, and IFN-γ was analyzed. The results displayed that mice that received the rHcARF1+PLGA NP, rHcARF1+CS NP, and antigen alone (rHcARF1) secreted a higher concentration of IL-4 and IL-12 as compared to control groups (Figure 4A). The opposite trend was observed in the case of IFN-γ (Figure 4D), which presented that rHcARF1 might play a critical role in producing specific cytokines and inducing the immunogenic response characterized by the differentiation of Th1/Th2 and immune cells cross-talk. Studies revealed that IL-17 could also significantly stimulate monocytes and DC to produce pro-inflammatory cytokines [70]. In addition, Th17 cells were reported to be involved in adaptive immunity to other pathogens [71]. In recent research, it was documented that the production of IL-17 was higher in those mice that received rHcARF1+PLGA NP compared to control groups (Figure 4C). This augmentation of IL-17 indicated that rHcARF1 might be involved in inducing a protective role against *H. contortus.*

TGF-β is a potent anti-inflammatory cytokine that acts on many target cells and tones down the inflammatory effects. It may serve as a potent suppressor of both Th1 and Th2 cells but foments the functions of T.reg cells [72]. However, a non-significant difference was observed in the secretion of TGF-β among the vaccinated and unvaccinated mice (Figure 4E). Hence, it is suggested that rHcARF1 supported the humoral immunity followed by the regulation of several biological processes. Th1 cells are typically essential in the clearance of intracellular pathogens, whereas Th2 cells are commonly associated with responses to parasitic infections. Segregation into Th1 or Th2 cells is dependent primarily on the cytokines released by APC [73]. Many genetically engineered mice (GEM) are backcrossed into a C57BL/6 genetic background. While C57BL/6 mice have been demonstrated to have a Th1-type bias to pathogens, whereas mice of other backgrounds, such as BALB/c tend toward a Th2-predominant response [74]. Gadahi et al. reported that HcARF1 stimulated PBMC and induced secretions of Th2 cytokines in the host animal [5]. However, Th1 or Th2 polarization could be a mouse strain-specific feature and not just due to the antigen. The skewing effect of PLGA+rHcARF1 on Th1 or Th2 polarization can only be confirmed with different mouse strains that need to be further probed.

The induction of humoral immunity is also the main priority for many vaccines, and humoral immune response controls extracellular pathogens through antibody binding and neutralization [20,75]. In mice, IgG2a produced from Th1 cells indicates cell-mediated immunity, and IgG1 produced from Th2 cells indicates humoral immunity. The results showed that rHcARF1 is immunogenic and agrees with the theory that virtually any parasite protein may act as an antigen, regardless of its location in the parasite [76]. The finding that PLGA NP containing rHcARF1 induces a more robust response toward humoral immunity than rHcARF1 alone in PBS and other control groups is logical. A significant difference (*** *p* < 0.001) was observed in the amount of IgG1, IgG2a, and IgM in comparison to control groups (Figure 3). Furthermore, the particle size of NP was smaller (Table 2), and smaller spherical particles generate a strong immune response [77]. Therefore, adequate initial antigen exposure, more extended antigen duration release properties (Figure 2B), and a smaller particle size of PLGA and CS NP (Table 2) could induce a more robust humoral immune response.

Lymphocyte proliferation is an essential scale that measures immunity [36]. To evaluate the proliferation ability of splenocytes, we performed a lymphoproliferation assay. The enhanced value of the SI showed that the splenic lymphocytes were more evident in the case of the rHcARF1+PLGA NP group (Figure 5). After getting the results, it can be suggested that the rHcARF1+PLGA NP antigen delivery system might be the most potent way to cause the multiplication of lymphocytes.

Efficient adaptive immune responses are characterized by the stimulation of antigen-specific CD4^+^ and CD8^+^ T lymphocytes that proliferate and differentiate into effector T cells. It has been demonstrated that the NP uptake by DC, antigen processing, and presentation are essential parameters for the activation and differentiation of T cells [40]. Additionally, CD8^+^ T cells mediate their functions with a combination of CD4^+^ T cells [78]. The tested groups of vaccinated mice, specifically rHcARF1+PLGA, showed the highest percentages of CD4^+^ and CD8^+^ cells than control groups. The results also revealed that rHcARF1+PLGA NP has a strong ability in inducing both CD4^+^ and CD8^+^ T cells (Figure 6).

Dendritic cells (DC) are the most potent APC, making them one of the key players in the cross-talk between innate and adaptive immunity. Given their capacity to capture and process the internalized antigens, DC efficiently prime naive T cells against foreign antigens and polarize them toward distinct effector fates [79,80]. Moreover, the activation mode of specific DC subsets through antigen engulfment determines that clinically relevant responses are dominated by cytotoxic T cells [81]. In the current study, the percentages of CD11c^+^CD83^+^ cells and CD11c^+^CD86^+^ cells increased in the mice injected with nano-vaccine compared with control groups (Figure 7). The data proved the increased maturation of DC in mice that were vaccinated with rHcARF1+PLGA nanoparticles.

## 5. Conclusions

The current study investigated the impact of HcARF1 with two different formulations of biodegradable polymers, PLGA and CS, which are used as the adjuvants in mice. The results of the SEM indicated that the recombinant antigen encapsulated in tiny nanoparticles competently and induced effective immune response by producing significant quantities of serum antibody titers (IgG1, IgG2a, IgM) against *H. contortus* infection. The induction of DC maturation (CD83^+^, CD86^+^), the proliferation of T cells (CD4^+^, CD8^+^), and splenic rise lymphocytes proved the immunogenicity of the HcARF1 and adjuvant activity of NP. Based on these in vivo results, rHcARF1+PLGA demonstrated an intense immune-enhancement activity comparatively. Antigen-loaded NP might play an essential role in the development of the immune system of the host.

## Figures and Tables

**Figure 1 vaccines-08-00726-f001:**
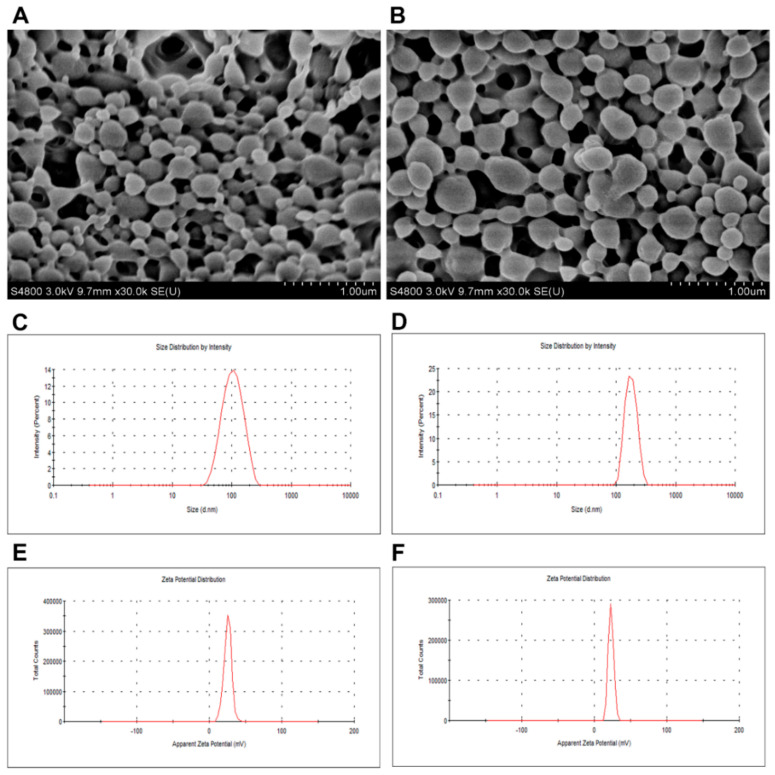
The morphology of nanoparticles was identified by scanning electron microscope (**A**,**B**). Particle size distribution (**C**,**D**) with a zeta potential of antigen-loaded poly ((**D**), L-lactide-co-glycolide) (PLGA) and chitosan (CS) NP (**E**,**F**) were also determined in the experiment.

**Figure 2 vaccines-08-00726-f002:**
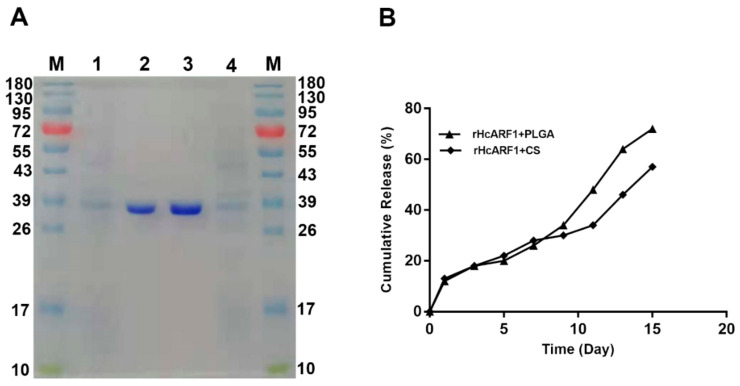
SDS-PAGE and cumulative release of antigen-loaded nanoparticles. SDS-PAGE of rHcARF1 entrapped in PLGA and CS NP. (**A**) Lane M: standard molecular weight protein marker, Lane 1: PLGA NP with unbounded rHcARF1. Lane 2: PLGA NP with bounded rHcARF1. Lane 3: CS NP with bounded rHcARF1. Lane 4: CS NPs with unbounded rHcARF1. (**B**) In vitro release profile of antigen from PLGA and CS NP at pH 7.4 at 37 °C for 14 days, calculated as a percentage release.

**Figure 3 vaccines-08-00726-f003:**
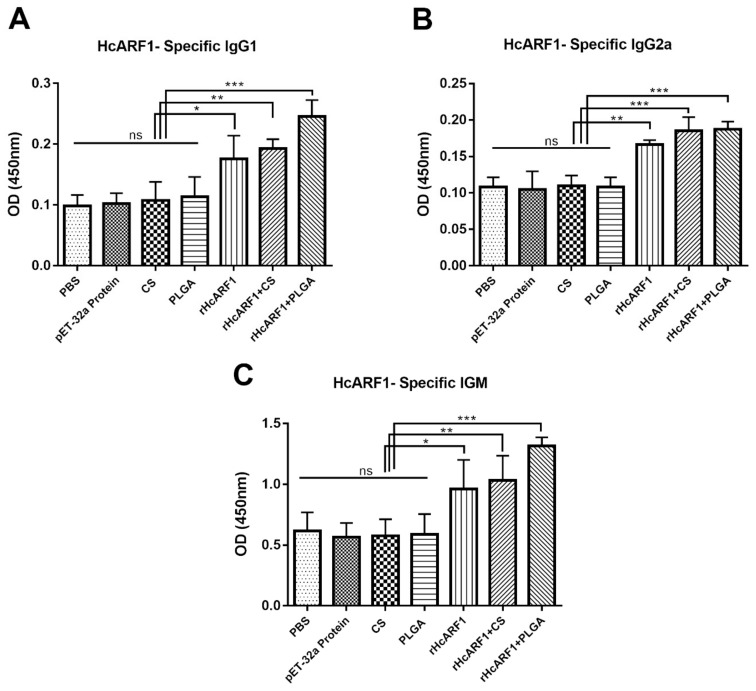
Effects of different antigen delivery systems on the antibody expression. The sera were collected from mice before sacrificing and detecting the antibodies secretion level changes by ELISA. Serum samples were collected on day 14 and the titers for IgG1 (**A**), IgG2a (**B**), and IgM (**C**) determined. Data are representative of triplicate experiments (* *p* < 0.05, ** *p* < 0.01, *** *p* < 0.001).

**Figure 4 vaccines-08-00726-f004:**
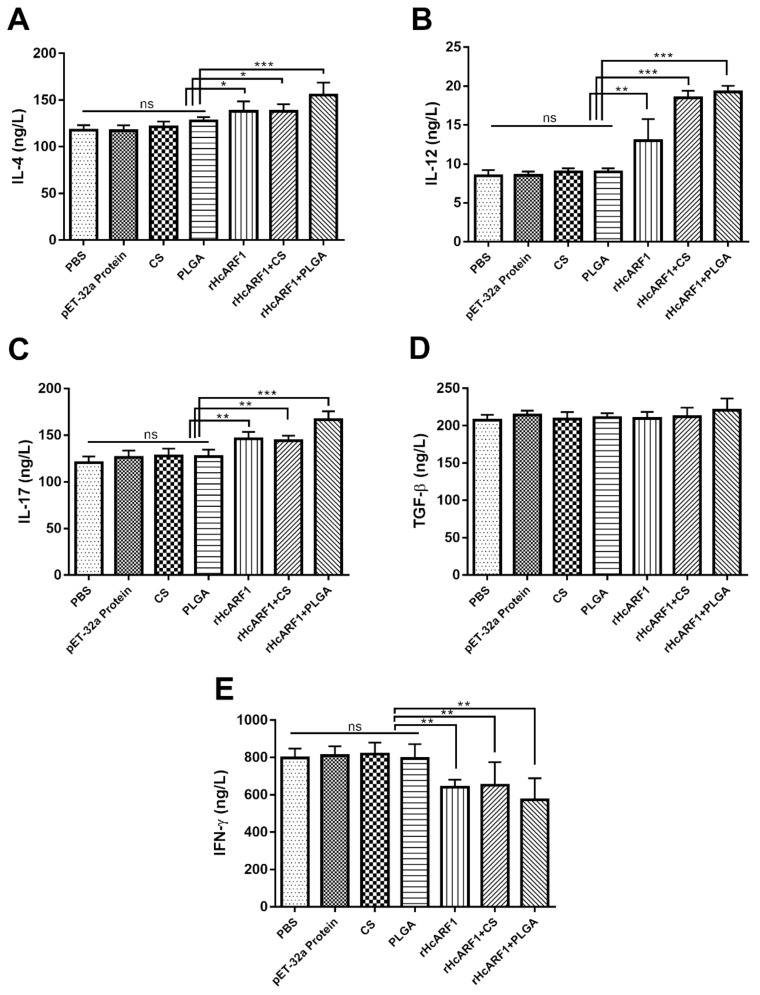
Effects of different antigen delivery systems on multiple cytokines expression. The sera were collected from mice before sacrificed and detected the cytokine released on the 14th day of experiment. Cytokine secretions for IL-4 (**A**), IL-12 (**B**), IL-17 (**C**), TGF-β1 (**D**) and IFN-γ (**E**) in the serum samples of mice quantified by ELISA. Data are representative of independent experiments triplicate in each (* *p* < 0.05, ** *p* < 0.01, *** *p* < 0.001).

**Figure 5 vaccines-08-00726-f005:**
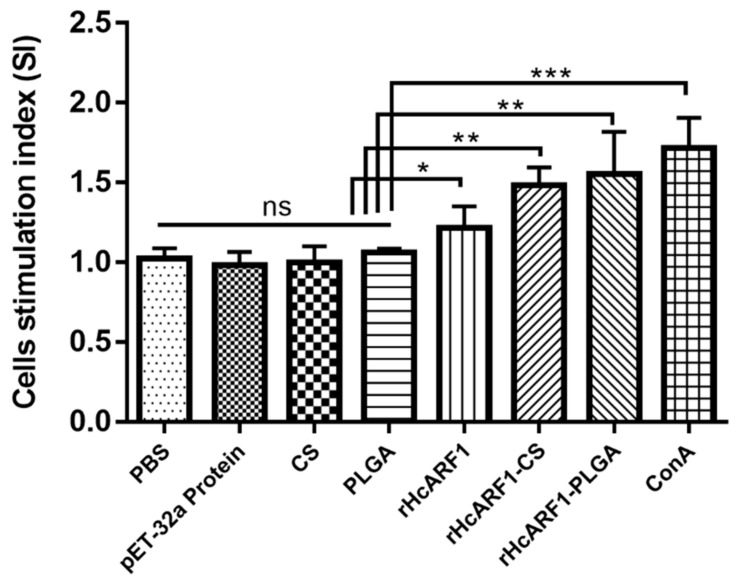
Lymphocyte proliferation was evaluated in all groups of mice by lymphocyte proliferation assay. The result was presented in the form of the stimulation index (SI). The data are representative of three independent experiments and the values presented here are the means ± SEM (* *p* < 0.05, ** *p* < 0.01, *** *p* < 0.001).

**Figure 6 vaccines-08-00726-f006:**
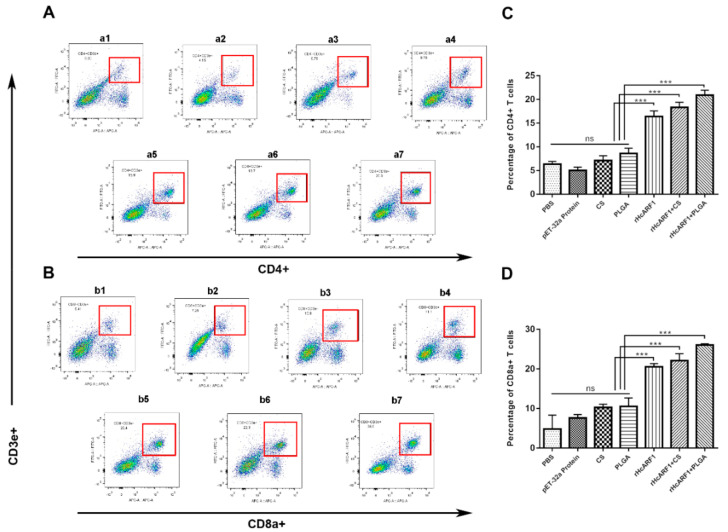
The proportions of CD4+ T cells and CD8+ T cells in all groups of mice affected by antigen-loaded nanoparticles. The percentages of CD4+ T cells ((**A**) (**a1**–**a7**)) and CD8+ T cells ((**B**) (**b1**–**b7**)) in seven groups could be shown as dot plots ((**a1**,**b1**): PBS group. (**a2**,**b2**): pET-32a protein group. (**a3**,**b3**): CS group. (**a4**,**b4**): PLGA group. (**a5**,**b5**): rHcARF1 group. (**a6**,**b6**): rHcARF1 + CS group. (**a7**,**b7**): rHcARF1+PLGA group, respectively determined by flow cytometry. The bar graphs show different treatments that affected the proportions of CD4+ T cells and CD8+ T cells (**C**,**D**). The results shown here are from an independent experiment that is representative of three independent experiments (*** *p* < 0.001).

**Figure 7 vaccines-08-00726-f007:**
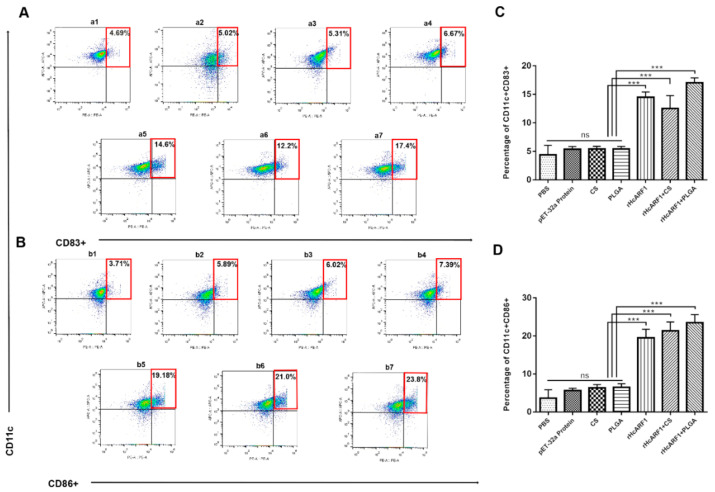
The effect of different antigen delivery systems on the splenic dendritic cell maturation. The expression of CD83^+^ ((**A**) (**a1**–**a7**)) and CD86^+^ ((**B**) (**b1**–**b7**)) in all experimental groups on splenic dendritic cells (DC) in seven groups could be shown as dot plots ((**a1**,**b1**): phosphate-buffered saline (PBS) group. (**a2**,**b2**): pET-32a protein group. (**a3**,**b3**): CS group. (**a4**,**b4**): PLGA group. (**a5**,**b5**): rHcARF1 group. (**a6**,**b6**): rHcARF1+CS group. (**a7**,**b7**): rHcARF1+PLGA group) were examined by flow cytometry. In (**C**,**D**), the bar graphs display different treatments that affected the proportion of CD83^+^ cells and CD86^+^ cells. The data are presented as the mean ± SEM and representative of triplicate experiments (*** *p* < 0.001).

**Table 1 vaccines-08-00726-t001:** Nature and composition of the different materials (vaccine) injected into mice and evaluated the type of immune response.

Groups	Inoculations	Injection at 0 Day	Purpose of Injection
1	PBS	1	Blank control
2	pET-32a Protein	1	Negative control
3	Chitosan (CS)	1	Immunogenicity of chitosan nanoparticles
4	PLGA	1	Immunogenicity of PLGA nanoparticles
5	rHcARF1	1	Immunogenicity of rHcARF1
6	rHcARF1-CS	1	Immunogenicity of rHcARF1 with chitosan nanoparticle
7	rHcARF1-PLGA	1	Immunogenicity of rHcARF1 with PLGA nanoparticle

**Table 2 vaccines-08-00726-t002:** Characterization of recombinant antigen (rHcARF1)-loaded PLGA and CS nanoparticles. Data are presented as the mean ± SD (*n* = 3).

NPs	Size (nm)	LC ^a^	EE ^b^	Zeta Potential (mV)
rHcARF1 + PLGA	100 ± 10	28.8	81.33	24 ± 1.8
rHcARF1 + CS	260 ± 35	40.6	76	18 ± 2.2

LC^a^ = (total protein − unbound protein)/total dry weight of Nano-vaccine × 100%. EE ^b^ = (total protein − unbound protein)/total protein × 100%.

## Data Availability

The datasets supporting the conclusions of this article are included in the report.

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
