# Peer review of "Nanoparticles (PLGA and Chitosan)-Entrapped ADP-Ribosylation Factor 1 of Haemonchus contortus Enhances the Immune Responses in ICR Mice"

_vaccines, 2020, doi:10.3390/vaccines8040726_

Round 1
Reviewer 1 Report
In the light of investigating an efficient immunogenic response for domesticated animals, the authors analyzed the impact of HcARF1 with two different formulations of biodegradable polymers (poly (D, L-lactide-co-glycolide), PLGA, and 19 chitosan (CS) in nanoparticles (NP) used as the adjuvants in mice.
My comments:
Introduction is pertinent and well summarizes the topic.
In Experimentals immunization protocol was correctly performed, with the due controls.
The employed methodologies were accurate and corresponding to the purpose of the experiments.
Statistic analysis allowed to evaluate the significance of data particularly from Figure 3 to 6. CD4+ and CD8+ T cells stimulation and induction of DC phenotypes were well highlighted by flow cytometry.
The induction of DC maturation (CD83+, CD86+), the proliferation of T cells (CD4+, 494 CD8+), and splenic rise lymphocytes adequately proved the immunogenicity of the HcARF1 and nanoparticles, demonstrating an intense immune-enhancement activity.
References are sufficiently updated.
Unless editing some errors and minor points reported below, the article is acceptable for publication in Vaccines.
Minor points
- 2.3 section-is there a specific reason to select females? Specify it in he text.
- 2.4 sect.-Did authors check the purification level of proteins by electrophoresis? It would be helpful for reader to see the gel figure.
- 2.5sect.- lines 121 and 130, 40,000 rpm, better espressed in g, here and followings.
line 3: Haemonchus contortus > Haemonchus contortus
line 101: one mM EDTA and one mM PSMF > 1 mM EDTA and 1 mM PSMF
line 209: Concanavalin A 12pt> 10pt
lines 264-265: please check that legend description corresponds to the lane numbers of panel A.
REf 1- 3: Haemonchus contortus > Haemonchus contortus
REf 1-2: year is not in bold. Checkalsothe followins (Ref. 11-12, 14, 19, 21, 22, 45, 46, 52, 54, 57, 60, 61, 64).
Author Response
Reviewer 1
In the light of investigating an efficient immunogenic response for domesticated animals, the authors analyzed the impact of HcARF1 with two different formulations of biodegradable polymers (poly (D, L-lactide-co-glycolide), PLGA, and 19 chitosan (CS) in nanoparticles (NP) used as the adjuvants in mice.
My comments:
Introduction is pertinent and well summarizes the topic.
In Experimentals immunization protocol was correctly performed, with the due controls.
The employed methodologies were accurate and corresponding to the purpose of the experiments.
Statistic analysis allowed to evaluate the significance of data particularly from Figure 3 to 6. CD4+ and CD8+ T cells stimulation and induction of DC phenotypes were well highlighted by flow cytometry.
The induction of DC maturation (CD83+, CD86+), the proliferation of T cells (CD4+, 494 CD8+), and splenic rise lymphocytes adequately proved the immunogenicity of the HcARF1 and nanoparticles, demonstrating an intense immune-enhancement activity.
References are sufficiently updated.
Unless editing some errors and minor points reported below, the article is acceptable for publication in Vaccines.
Minor points
- 2.3 section-is there a specific reason to select females? Specify it in the text.
Response:
Thank you very much for your suggestion. We have corrected the mistake described in Line 92-93 of revised manuscript.
- 2.4 sect.-Did authors check the purification level of proteins by electrophoresis? It would be helpful for reader to see the gel figure.
Response:
Thank you very much for your suggestion. We have checked and corrected in revised manuscript, described in Line 107-110.
- 2.5sect.- lines 121 and 130, 40,000 rpm, better espressed in g, here and followings.
Response:
Thank you very much for your suggestion. We have corrected the mistake described in Line 125, 134 of section 2.5 in revised manuscript.
line 3: Haemonchus contortus > Haemonchus contortus
Response:
Thank you very much for your suggestion. We have corrected the mistake, described in title of revised manuscript.
line 101: one mM EDTA and one mM PSMF > 1 mM EDTA and 1 mM PSMF
Response:
Thank you very much for your suggestion. We have corrected the mistake, described in Line 103 of revised manuscript.
line 209: Concanavalin A 12pt> 10pt
Response:
Thank you very much for your suggestion. We have corrected the mistake, described in Line 215 of revised manuscript.
lines 264-265: please check that legend description corresponds to the lane numbers of panel A.
Response:
Thank you very much for your suggestion. We have corrected the mistake, described in Line 270-271 of revised manuscript.
REf 1- 3: Haemonchus contortus > Haemonchus contortus
Response:
Thank you very much for your suggestion. We have checked and corrected the mistake in references of revised manuscript.
REf 1-2: year is not in bold. Check also the followins (Ref. 11-12, 14, 19, 21, 22, 45, 46, 52, 54, 57, 60, 61, 64).
Response:
Thank you very much for your suggestion. We have checked and corrected the mistake in references of revised manuscript accordingly.
Submission Date
18 October 2020
Date of this review
27 Oct 2020 17:14:08

Reviewer 2 Report
In the manuscript entitled, “Nanoparticles (PLGA and chitosan) entrapped ADP-ribosylation Factor 1 of Haemonchus contortus enhance the immune responses in a model animal of mice.” Hasan et al have used nanoparticle entrapment to improve vaccine response against Haemonchus contortus HcARF1 protein. I have one comment to improve the readability of the manuscript.
- In figures 3 – 8, it is difficult to understand which sample is being compared to what. For example, a bar is used to indicate the two samples being compared. However, in figure 3b, it appears that authors are comparing rHcARF1 + CS with rHcARF1 + PLGA. Looking at the data there is now way that there is a statistically significant difference between them.
Author Response
Reviewer 2
In the manuscript entitled, “Nanoparticles (PLGA and chitosan) entrapped ADP-ribosylation Factor 1 of Haemonchus contortus enhance the immune responses in a model animal of mice.” Hasan et al have used nanoparticle entrapment to improve vaccine response against Haemonchus contortus HcARF1 protein. I have one comment to improve the readability of the manuscript.
- In figures 3 – 8, it is difficult to understand which sample is being compared to what. For example, a bar is used to indicate the two samples being compared. However, in figure 3b, it appears that authors are comparing rHcARF1 + CS with rHcARF1 + PLGA. Looking at the data there is now way that there is a statistically significant difference between them.
Response:
Thank you very much for your suggestion. We have edited and changed the figures # 3-8 accordingly in the revised manuscript.
Submission Date
18 October 2020
Date of this review
26 Oct 2020 19:49:15

Reviewer 3 Report
Major concerns:
The manuscript focuses on developing a nanoparticle-loaded ARF1 immunogen against H. contortus in mouse model. The authors have analyzed the overall changes in the immune parameters in control and immunized mice and concluded that ARF1-loaded NPs can be effective against the parasite. However, several key approaches are missing that would have enabled us to understand whether this vaccine formulation has any biological relevance.
It can be said that the important focus of any immunization study is to see whether an antigen-specific immunity is stimulated that offers a protective response against the parasitic infestation. No such experiments are attempted here. One could argue that any foreign protein when encapsulated in nanoparticles, as done in this study, and properly immunized is going to evoke similar immunological changes in mice. Thus, the real concept of induction of antigen-specific protective immunity is not addressed here.
- Authors claim to have measured antigen-specific antibody/cytokine response (lines 24, 181, 195, 281, 284 etc), instead their ELISA protocols explains only measuring the total antibody, regardless of antigen specificity. For example, ARF-specific antibodies would be measured by coating ARF1 in ELISA wells (not anti-isotype mAb, as given by the authors) and the resulting serum binding be detected with specific isotype-recognizing detection antibodies. Otherwise, they are only measuring a total increase in the antibody titer.
- The Th1/Th2 ratio (Figure 4) measured by the authors needs a bit more explanation methods-wise, as it is very confusing. Were the antibody titers shown in Figure 3A and 3B used for this calculation? Mere eye-balling of Figure 3 would tell that the ratio of IgG2a vs IgG1 levels (ng/mL) should be more than 3 (e.g for PBS and controls). However, Figure 4 shows a ratio as <1 for PBS and control mice. Please explain the methods used to arrive at this data?
- A higher IgG2a may mean a Th1-polarizing response, as discussed by the authors (Figure 4); however, the cytokine profile (Figure 5) seems to be opposite of this hypothesis. In fact, the changes in cytokine in Figure 4 would mean a polarization towards Th2 and Th17, but not Th1. Too little data is shown/measured to define an immune response as either Th1 or Th2. Such definition may need additional T cell expression signatures, which is not shown in this manuscript.
- Lymphocyte stimulation was performed using ConA-based stimulation. As ConA is a general mitogen for T cells, one would expect a polyclonal T cell proliferation and not antigen-specific activity. Such approach does not validate whether the T cell response generated by the vaccine is antigen-specific or non-specific. Instead it only tells us that there were more T cells in the spleen that could be activated by ConA. Antigen-specific T cell proliferation should be measured by using antigen-pulsed DCs for stimulation.
- Presence of antigen-specific lymphocytes in the draining lymph nodes at the immunization site (done at earlier time point) should be shown to indicate that the changes follow the kinetics of immunization.
- Protective response of the ARF1 is not shown. This is an essential experiment that is needed for a complete immunization study. Was the nanoparticle-loaded ARF1 able to significantly reduce parasitic/egg load in the immunized animals when compared to the control mice?
- The route of immunization can be specified in the Methods section, as well.
Author Response
Reviewer 3
Major concerns:
The manuscript focuses on developing a nanoparticle-loaded ARF1 immunogen against H. contortus in mouse model. The authors have analyzed the overall changes in the immune parameters in control and immunized mice and concluded that ARF1-loaded NPs can be effective against the parasite. However, several key approaches are missing that would have enabled us to understand whether this vaccine formulation has any biological relevance.
It can be said that the important focus of any immunization study is to see whether an antigen-specific immunity is stimulated that offers a protective response against the parasitic infestation. No such experiments are attempted here. One could argue that any foreign protein when encapsulated in nanoparticles, as done in this study, and properly immunized is going to evoke similar immunological changes in mice. Thus, the real concept of induction of antigen-specific protective immunity is not addressed here.
- Authors claim to have measured antigen-specific antibody/cytokine response (lines 24, 181, 195, 281, 284 etc), instead their ELISA protocols explains only measuring the total antibody, regardless of antigen specificity. For example, ARF-specific antibodies would be measured by coating ARF1 in ELISA wells (not anti-isotype mAb, as given by the authors) and the resulting serum binding be detected with specific isotype-recognizing detection antibodies. Otherwise, they are only measuring a total increase in the antibody titer.
Response:
Thank you very much for your suggestion. We have checked and corrected the mistake, described in Line 187-193 in revised manuscript.
- The Th1/Th2 ratio (Figure 4) measured by the authors needs a bit more explanation methods-wise, as it is very confusing. Were the antibody titers shown in Figure 3A and 3B used for this calculation? Mere eye-balling of Figure 3 would tell that the ratio of IgG2a vs IgG1 levels (ng/mL) should be more than 3 (e.g for PBS and controls). However, Figure 4 shows a ratio as <1 for PBS and control mice. Please explain the methods used to arrive at this data?
Response:
Thank you very much for your question. We have corrected the mentioned mistake, described in Line 306-307 of main text and figure 4 have edited also in revised manuscript.
- A higher IgG2a may mean a Th1-polarizing response, as discussed by the authors (Figure 4); however, the cytokine profile (Figure 5) seems to be opposite of this hypothesis. In fact, the changes in cytokine in Figure 4 would mean a polarization towards Th2 and Th17, but not Th1. Too little data is shown/measured to define an immune response as either Th1 or Th2. Such definition may need additional T cell expression signatures, which is not shown in this manuscript.
Response:
Thank you very much for your query. As reported in previous study, HcARF1 induce Th2 cytokines while incubated with goat PBMCs. Similarly, we found increased secretion of Th2 cytokines in recent experiment. Moreover, the main effector cells of Th1 immunity are CD8 T cells. In Figure 7, we reported that HcARF1 stimulate CD4, and CD8 T Cells. Therefore, HcARF1 is responsible for inducing both Th1 and Th2 immune response in host animal.
Javaid Ali Gadahi, Muhammad Ehsan, Shuai Wang, Zhenchao Zhang, Ruofeng Yan, Xiaokai Song, Lixin Xu and Xiangrui Li. Recombinant protein of Haemonchus contortus small GTPase ADP-ribosylation factor 1 (HcARF1) modulate the cell mediated immune response in vitro. Oncotarget, 2017, Vol. 8, (No. 68), pp: 112211-112221.
Alba Cortés, Carla Muñoz-Antoli, J. Guillermo Esteban, and Rafael Toledo. Th2 and Th1 Responses: Clear and Hidden Sides of Immunity Against Intestinal Helminths. 2017. DOI:https://doi.org/10.1016/j.pt.2017.05.004
- Lymphocyte stimulation was performed using ConA-based stimulation. As ConA is a general mitogen for T cells, one would expect a polyclonal T cell proliferation and not antigen-specific activity. Such approach does not validate whether the T cell response generated by the vaccine is antigen-specific or non-specific. Instead it only tells us that there were more T cells in the spleen that could be activated by ConA. Antigen-specific T cell proliferation should be measured by using antigen-pulsed DCs for stimulation.
Response:
Thank you very much for this query. Authors used the ConA as it is reported in many previous studies in experiments stated T cells proliferation.
Lijia Yuan, Yong Wang, Xiaodan Ma, Xuemei Cui, Meiqian Lu, Ran Guan, Xiaoqing Chi, Wei Xu, Songhua Hu. Sunflower seed oil combined with ginseng stem-leaf saponins as an adjuvant to enhance the immune response elicited by Newcastle disease vaccine in chickens. Vaccine 38 (2020) 5343–5354.
Nian-Zhang Zhang1†, Ying Xu1,2†, Meng Wang1, Jia Chen1, Si-Yang Huang1,4, Qi Gao3 and Xing-Quan Zhu. Vaccination with Toxoplasma gondii calcium-dependent protein kinase 6 and rhoptry protein 18 encapsulated in poly(lactide-co-glycolide) microspheres induces long-term protective immunity in mice. Zhang et al. BMC Infectious Diseases (2016) 16:168 DOI 10.1186/s12879-016-1496-0.
- Presence of antigen-specific lymphocytes in the draining lymph nodes at the immunization site (done at earlier time point) should be shown to indicate that the changes follow the kinetics of immunization.
Response:
Thank you very much for your suggestion. We would consider it in our future publication.
- Protective response of the ARF1 is not shown. This is an essential experiment that is needed for a complete immunization study. Was the nanoparticle-loaded ARF1 able to significantly reduce parasitic/egg load in the immunized animals when compared to the control mice?
Response:
Thank you very much for your suggestion. We discussed this aspect of protective response induced by HcARF1 in our other publication submitted in other journal in which we described in detail the reducing effect of male/female/total parasite count.
Muhammad Waqqas Hasan, Qiang Qiang Wang, Muhammad Ehsan, Javaid Ali Gadahi, Muhammad Ali-ul-Husnain Naqvi, Tahir Aleem, Haider Ali, Shakeel Ahmed Lakho, Ruo Feng Yan, Li Xin Xu, Xiao Kai Song, Xiangrui Li. Recombinant HCA59 and ARF1 of Haemonchus contortus incorporated in PLGA nanoparticles mediate the immunoglobulins (IgG, IgA, IgE) secretion elicit the protective immunity in goats. (Submitted)
- The route of immunization can be specified in the Methods section, as well.
Response:
Thank you very much for your question. We have already described it. Please read Line 174 of revised manuscript.
Submission Date
18 October 2020
Date of this review
05 Nov 2020 23:35:12

Round 2
Reviewer 3 Report
Concerns:
- New description in Line 188 about the ELISA is confusing. Were the plates coated with recombinant ARF1 or with anti-mouse IgG1/2a/M antibodies or both? They are supposed to yield different types of data.
- Line 306 and Figure 4: Data shown in this figure is not convincing enough to conclude that there is a Th1 polarization induced by the antigen (as mentioned in line 453), but being discussed in many places in the text as if the ARF1 induced a Th1 polarization (e.g. line 453 and see next point).
- In fact, even in the non-immunized mice, the Th1/Th2 ratio is more than 1. This implies that the higher IgG2a/IgG1 ratio, and implied Th1 polarization, could be a mouse strain-specific feature and not just due to immunization. The skewing effect of PLGA-ARF1 on Th1 polarization, as determined based on IgG2a/G1 ratio can only be confirmed with different mouse strains. The authors may decide to delete Figure 4 and description of Th1 polarization effect by PLGA-ARF1 (section 3.4 and texts in discussion section), as it is not supported by their data and does not bring any significance to the manuscript.
- Line 445: The description that "rHcARF1 is majorly responsible for Th1 response" not supported by IgG or cytokine data (Figure 4 and 5), instead, it suggests a Th2-type (Figure 5). Th cell polarization is a complex feature of immune response. A broad panel of gene expression, including cytokines and transcription factors, would be needed for such studies. Authors also mention in their letter of response to the reviewer that ARF1 antigen could induce both Th1 and Th2-type of immune response. Better clarity in the discussion is required on this aspect to make sure the manuscript conveys the correct inference that is supported by data.
- Section 2.12: To measure rHcARF1-specific lymphocyte activation, was rHcARF1 protein used to restimulate the lymphocytes in culture or was ConA used as a restimulant in all the groups? It is not clear from their description. If rHcARF1 was used as a restimulation, then it would denote an antigen-specific response. On the other hand, if ConA was used as a restimulant, then the authors need to explicitly specify that the response noticed mainly includes a polyclonal and poly-specific stimulation.
- Line 456: No data has been provided to support this description about PLGA-NP containing rHcARF1 induced parasite eradication.
- The immune responses shown for PLGA or CS-encapsulated ARF1 appears to be very modest overall in magnitude when compared to ARF1 alone (though significant). It appears that ARF1 is already immunogenic in these mice, So, what is the advantage here for encapsulation in PLGA or CS? Did they actually improve antigen presentation or phagocytosis of particles or is the immune response long-lasting? Without comparing these, it is hard to appreciate the importance of PLGA/CS encapsulations. A discussion on this should be included in the discussion.
Author Response
Author’s Response
To,
The Editorial office
Vaccines
Nov 19, 2020
Manuscript ID: vaccines-987223  
We are pleased to resubmit our revised manuscript entitled: “Nanoparticles (PLGA and chitosan) entrapped ADP-ribosylation Factor 1 of Haemonchus contortus enhance the immune responses in a model animal of mice”.
Thank you very much for consideration of our manuscript (RVSC-D-20-00286). We have revised our manuscript thoroughly and considered all the issues mentioned in the editor and reviewer’s comments. The changed parts are highlighted as Track Changes in the revised manuscript. We hope that you and reviewers will be satisfied with the revised version of our document.
Open Review
(x) I would not like to sign my review report
( ) I would like to sign my review report
English language and style
( ) Extensive editing of English language and style required
( ) Moderate English changes required
(x) English language and style are fine/minor spell check required
( ) I don't feel qualified to judge about the English language and style
Yes |
Can be improved |
Must be improved |
Not applicable |
|
Does the introduction provide sufficient background and include all relevant references? |
(x) |
( ) |
( ) |
( ) |
Is the research design appropriate? |
( ) |
( ) |
(x) |
( ) |
Are the methods adequately described? |
( ) |
( ) |
(x) |
( ) |
Are the results clearly presented? |
( ) |
(x) |
( ) |
( ) |
Are the conclusions supported by the results? |
( ) |
( ) |
(x) |
( ) |
Comments and Suggestions for Authors
Concerns:
- New description in Line 188 about the ELISA is confusing. Were the plates coated with recombinant ARF1 or with anti-mouse IgG1/2a/M antibodies or both? They are supposed to yield different types of data.
Response:
Thank you very much for your attention. We have revised the methodology mentioned in section 2.9, and corrected the confusing statement, and the changes are highlighted in Line 192-198 of revised manuscript.
- Line 306 and Figure 4: Data shown in this figure is not convincing enough to conclude that there is a Th1 polarization induced by the antigen (as mentioned in line 453), but being discussed in many places in the text as if the ARF1 induced a Th1 polarization (e.g. line 453 and see next point).
- In fact, even in the non-immunized mice, the Th1/Th2 ratio is more than 1. This implies that the higher IgG2a/IgG1 ratio, and implied Th1 polarization, could be a mouse strain-specific feature and not just due to immunization. The skewing effect of PLGA-ARF1 on Th1 polarization, as determined based on IgG2a/G1 ratio can only be confirmed with different mouse strains. The authors may decide to delete Figure 4 and description of Th1 polarization effect by PLGA-ARF1 (section 3.4 and texts in discussion section), as it is
- Line 445: The description that "rHcARF1 is majorly responsible for Th1 response" not supported by IgG or cytokine data (Figure 4 and 5), instead, it suggests a Th2-type (Figure 5). The cell polarization is a complex feature of immune response. A broad panel of gene expression, including cytokines and transcription factors, would be needed for such studies. Authors also mention in their letter of response to the reviewer that ARF1 antigen could induce both Th1 and Th2-type of immune response. Better clarity in the discussion is required on this aspect to make sure the manuscript conveys the correct inference that is supported by data.
Response:
Thank you very much for your attention. We have deleted the Figure 4 from the main text as above mentioned. Moreover, we have added some explanation about Th1 or Th2 immune response, described in Line 427-430, and 448-458 of revised manuscript.
- Section 2.12: To measure rHcARF1-specific lymphocyte activation, was rHcARF1 protein used to restimulate the lymphocytes in culture or was ConA used as a restimulant in all the groups? It is not clear from their description. If rHcARF1 was used as a restimulation, then it would denote an antigen-specific response. On the other hand, if ConA was used as a restimulant, then the authors need to explicitly specify that the response noticed mainly includes a polyclonal and poly-specific stimulation.
Response:
Thank you very much for your attention. We have corrected the mistake, described in Line 215-216 of revised manuscript.
- Line 456: No data has been provided to support this description about PLGA-NP containing rHcARF1 induced parasite eradication.
Response:
Thank you very much for your attention. We have deleted the statement which does not support the conclusion of our study.
- The immune responses shown for PLGA or CS-encapsulated ARF1 appears to be very modest overall in magnitude when compared to ARF1 alone (though significant). It appears that ARF1 is already immunogenic in these mice, So, what is the advantage here for encapsulation in PLGA or CS? Did they actually improve antigen presentation or phagocytosis of particles or is the immune response long-lasting? Without comparing these, it is hard to appreciate the importance of PLGA/CS encapsulations. A discussion on this should be included in the discussion.
Response:
Thank you very much for your attention. Authors have corrected the mistake, described in Lines 408-420 of revised manuscript. Additionally, we have added some lines about the importance of ARFs in introduction part of revised manuscript, describes in Lines 46-51
